# Analysis of Polar Lipids in Hemp (*Cannabis sativa* L.) By-Products by Ultra-High Performance Liquid Chromatography and High-Resolution Mass Spectrometry

**DOI:** 10.3390/molecules27185856

**Published:** 2022-09-09

**Authors:** Arjun H. Banskota, Alysson Jones, Joseph P. M. Hui, Roumiana Stefanova, Ian W. Burton

**Affiliations:** Aquatic and Crop Resource Development Research Centre, National Research Council Canada, 1411 Oxford Street, Halifax, NS B3H 3Z1, Canada

**Keywords:** hemp, diacylglycerols, phospholipids, phosphatidylcholine, lysophosphatidylcholine, lysophosphatidylethanolamine, phosphatidylethanolamine, *Cannabis sativa*, hemp seed hulls, hemp cake

## Abstract

Polar lipids were extracted from residual biomass of hemp (*Cannabis sativa* L.) by-products with EtOH and partitioned into aqueous and chloroform fractions. The chloroform fractions were studied for their lipid composition using solid-phase extraction (SPE) followed by UHPLC/HRMS and NMR analyses. The ^1^H NMR and gravimetric yield of SPE indicated triacylglycerols covered ≥ 51.3% of the chloroform fraction of hemp seed hulls and hemp cake. UHPLC/HRMS analyses of remaining polar lipids led to the identification of nine diacylglycerols (DAGs), six lysophosphatidylcholines (LPCs), five lysophosphatidylethanolamines (LPEs), eight phosphatidylethanolamines (PEs), and thirteen phosphatidylcholines (PCs) for the first time from hemp seed hulls. The regiospecificity of fatty acyl substitutes in glycerol backbone of individual phospholipids were assigned by analyzing the diagnostic fragment ions and their intensities. The heat-map analysis suggested that DAG 18:2/18:2, 1-LPC 18:2, 1-LPE 18:2, PE 18:2/18:2, and PC 18:2/18:2 were the predominant molecules within their classes, supported by the fact that linoleic acid was the major fatty acid covering > 41.1% of the total fatty acids determined by GC-FID analysis. The ^31^P NMR analysis confirmed the identification of phospholipids and suggested PC covers ≥ 37.9% of the total phospholipid present in hemp by-products. HPLC purification led to the isolation of 1,2-dilinoleoylphosphatidylcholine and 1-palmitoyl-2-linoleoylphosphatidylcholine. These two major PCs further confirmed the UHPLC/HRMS finding.

## 1. Introduction

Industrial hemp (*Cannabis sativa* L.) is a part of the Cannabis family and contains low levels of delta-9-tetrahydrocannabinol (Δ-9-THC), a psychoactive molecule found in marijuana [1]. Hemp has been grown worldwide for food or production of other household products, such as paper, textiles, fibers, insulation, and biofuel [2]. Hemp is emerging as a potential source for high-value functional food ingredients and nutraceuticals [3], because of its excellent polyunsaturated fatty acid (PUFA) profile and high protein content [4,5,6,7]. Hemp extract is also reported to have antimicrobial properties [8], and hemp seeds exhibit antioxidant property and cytotoxic activity against NCI-H460 cells [9]. Industrial hemp is grown worldwide in both temperate and tropical climates. In Canada, it is grown mainly in the prairies (Alberta, Saskatchewan, and Manitoba). According to Health Canada, 77,800 acres of industrial hemp were planted in 2018, and this is expected to increase in coming years [10]. Identification of high-value chemicals from hemp seed by-products may provide an additional revenue stream for this emerging agroindustry.

Polar lipids including glycerolipids (GL), glycerophospholipids (GP) and sphingolipids (SP) are part of membrane lipids that also act as important signaling molecules between cells. They have biological significance and are reported to have interesting biological activities. GLs, especially sulfoquinovosyldiacylglycerols (SQDGs), are reported to have potent inhibitors of eukaryotic DNA polymerases and HIV reverse transcriptase type 1 [11]. Aquatic-source GPs have a substantial amount of *n*-3 polyunsaturated fatty acids (PUFAs), particularly eicosapentaenoic acid (EPA) and docosahexaenoic acid (DHA). GPs have also been reported to show antioxidant activity, improving immunity, and preventing cardiovascular diseases [12]. Polar lipids, especially galactolipids, isolated from medicinal plants and green vegetables, algae, and bacteria are reported to have anti-inflammatory and anticancer activities [13]. With regard to hemp lipids, a number of studies have been done on the characterization of lipid components using GC-FID, GC/MS, and LC/MS. Delgado-Povedano et al. (2020) [14] reported the identification of several cannabinoids, terpenoids, lipids and flavonoids when studying extracts from 17 cultivars of *C. sativa* by untargeted analysis. Cerrato et al. (2021) [15], on the other hand, outlined the analytical workflow for lipid characterization to identify several polar lipids. Similarly, Arena et al. (2022) [16] studied the lipid composition of hemp products by GC and UHPLC/MS, characterizing fatty acids and triacylglycerols (TAGs). In our previous study, we also reported the characterization of lipids present in oil and protein extracted from both hemp seeds and hemp seed by-products [17,18]. To continue our research on hemp by-product valorization, we further studied the polar lipids in hemp by-products, especially hemp seed hulls and hemp cake. We would like to report herein the identification of a number of diacylglycerols (DAGs), lysophosphatidylcholines (LPCs), lysophosphatidylethanolamines (LPEs), phosphatidylethanolamines (PEs), and phosphatidylcholines (PCs) present in the polar lipid fractions of hemp seed by-products and the purification of two major phospholipids i.e., 1,2-dilinoleoylphosphatidylcholine and 1-palmitoyl-2-linoleoylphosphatidylcholine.

## 2. Results

### 2.1. Extraction, Fractionation, and Solid-Phase Extraction (SPE)

The residual biomass of hemp cake and hemp seed hulls recovered after removal of oil (hexane extraction) were further extracted with EtOH at 60 °C. The EtOH extract was fractionated into chloroform- and water-soluble parts, and the yields of chloroform-soluble fractions were 1.5% and 1.6% for hemp cake and hemp seed hull biomass (before removing oil), respectively. The chloroform-soluble part of the EtOH extract was further divided into three subfractions using silica gel-based solid-phase extraction (SPE) cartridge eluting with chloroform followed by acetone and methanol. Percentages of individual subfractions are shown in Table 1. The fraction eluted with chloroform covered more than 51% for both hemp seed hulls and hemp cake. The acetone fraction was 22.1–24.7% and the MeOH fraction was 11.3–26.4% with respect to the chloroform-soluble part of EtOH extract.

### 2.2. H NMR Analysis of SPE Fractions

The ^1^H NMR spectra of subfractions obtained after SPE were recorded in deuterated solvents. The ^1^H NMR spectra of SPE fractions from hemp seed hulls are shown in Figure 1 and hemp cake are shown in Appendix A. The ^1^H NMR spectrum of each eluent of hemp seed hulls and hemp cake showed almost identical proton NMR signals (Figure 1 and Appendix A). The ^1^H NMR signals of the chloroform fraction are mainly from triacylglycerols (TAGs) [17]. The ^1^H NMR spectrum of the acetone fraction, on the other hand, showed strong signals at 6.00–8.00 ppm, corresponding to aromatic/conjugated double-bond protons. The unsaturated proton signals of 5.2–6.0 ppm and the aliphatic signals of 1.5–3.0 ppm clearly suggested the presence of a fatty acid moiety, indicating that the acetone fraction contained polar lipids. The methanol fraction also included the presence of fatty acid acyl chains (5.2–6.0 and 1.5–3.0 ppm), glycerol moiety (5.2–60 ppm), and additional methylene signals of 3.50–4.50 ppm.

### 2.3. Ultrahigh-Performance Liquid Chromatography and High-Resolution Mass Spectrometry (UHPLC/HRMS) Analysis

The total ion chromatograms (TICs) of acetone and methanol fractions of hemp seed hulls in positive mode are shown in Figure 2 and hemp cake are shown in Appendix A. The TICs of hemp cake and hemp seed hulls fractions in negative mode are shown in Appendix A, respectively. TICs of each eluent of hemp seed hulls and hemp cake showed close similarity in positive and negative modes, except the acetone fraction of hemp cake showed peaks between 0.5 and 1.0 min and were absent in the acetone fraction of hemp seed hulls. The major signals in the acetone fraction eluted between 7.00 and 8.00 min were from phytyl derivatives with *m*/*z* 871, 855, 887, 901, and 903. Eight diacylglycerols (DAGs) were identified eluting between 6.31 and 7.00 min (Table 2). Methanol fractions showed signals belonging to LPSs, LPEs, PEs, PCs, and a few unknown phospholipids (Table 3). LPSs were eluted between 2.10 and 5.00 min, PEs were eluted between 6.14 and 6.70 min, and PCs were eluted after 6.28 min. Identification of the individual lipids was made based on the accurate masses of the molecular adduct ions i.e., [M + H]^+^ or [M + NH_4_]^+^ and their corresponding fragmentation ions observed in both positive and negative modes.

### 2.4. Fatty Acid Analysis of SPE Fractions

The results of fatty acid analysis of all SPE fractions of hemp seed hulls and hemp cake are shown in Table 4. The results clearly demonstrated that linoleic acid (18:2 *n*-6) was the predominant fatty acid present in all the three SPE fractions, ranging from 41.1% to 57.2% of the total fatty acids, followed by α-linolenic acid (18:3 *n*-3), which ranged between 16.6% and 21.0% of the total fatty acid. The third most dominant fatty acid was palmitic acid (16:0). Overall PUFA compromised 61.0–76.4% of the total fatty acid. The chloroform fraction of both hemp by-products i.e., hemp seed hulls and hemp cake, had the highest total fatty acid content at 824.7 mg/g and 670.1 mg/g, respectively. Acetone fraction had the lowest total fatty acid content of <330.2 mg/g of total fatty acids.

### 2.5. P NMR and HPLC Purification of Methanol Fraction

The methanol fraction of both hemp by-products was further subjected to ^31^P NMR analysis. The ^31^P NMR spectra of both hemp seed hulls and hemp cake eluted with MeOH are shown in Figure 3, and suggested the presence of PC, 1-LPC, PE, and LPE in both samples. The relative percentage of each phospholipid is shown in Table 5. PC was the major phospholipid in MeOH fraction, covering >37% of total phospholipids. The purification of the major phospholipids 1,2-dilinoleoylphosphatidylcholine (PC 18:2/18:2) and 1-palmitoyl-2-linoleoylphosphatidylcholine (PC 16:0/18:2) were also achieved using HPLC and their structure confirmed by in-depth NMR analyses, including 1D- (^1^H and ^13^C) and 2D-NMR (COSY, HSQC, and HMBC) and HRMS spectra. The HPLC chromatogram is shown in Appendix A. The structure of 1,2-dilinoleoylphosphatidylcholine and ^1^H NMR spectrum with proton assignments are shown in Figure 4. The NMR spectra including 1D- (^1^H- and ^13^C-NMR) and 2D-NMR (COSY, HSQC, HMBC, and HRMS) of 1,2-dilinoleoylphosphatidylcholine and 1-palmitoyl-2-linoleoylphosphatidylcholine are presented in Appendix A.

## 3. Discussion

Hemp oil is reported to have an excellent polyunsaturated fatty acid profile containing high amounts of *α*-linolenic acid (*ω*-3) and linoleic acid (*ω*-6). These *ω*-3 and *ω*-6 fatty acids are mainly constituted in TAGs, and we previously reported the identification of 47 individual TAGs in cold-pressed commercial hemp oil and oil (hexane extract) extracted from either hemp seeds or from hemp seed by-products [17]. Recently Arena et al. (2022) also characterized the fatty acids and TAGs present in hemp seed oils and flours using GC-FID/MS and LC/MS, respectively [16]. During our study, we observed that hexane extracted 69.1% of the total lipid present in the hemp hulls. When hexane was used as an extraction solvent for hemp cake, only 56.4% of the lipids were extracted, mainly neutral lipids. These results clearly suggested that hemp by-products contain a significant amount of polar lipids [17].

Even though polar lipids, including glycolipids (GLs), phospholipids (PLs), and sphingolipids (SPs), are common components of seed oils, few studies have been done on polar lipid characterization of hemp seeds or hemp seed oils. Research on hemp seed or hemp seed oils is primarily focused on either extraction procedure/efficiency or fatty acid profile [19,20,21,22]. Ellison et al. (2021) [23] described the quantitative analysis of phospholipids from hemp seeds using near-critical CO_2_, propane, and dimethyl ether systems. The ^31^P NMR-based quantitative analysis demonstrated that hemp seed contains phosphatidylcholine (PC), phosphatidylinositol (PI), lysophosphatidylcholine (LPC), phosphatidylethanolamine (PE), and phosphatidic acid (PA). Among those groups, PC was reported to cover 40.3–42.8% of the total phospholipids, and none of the phospholipids was characterized or identified [23]. Buré et al. (2016) [24], on the other hand, studied the phospholipid composition of five plant cakes, including hemp, using electrospray mass spectrometry and identified several classes of phospholipid in hemp cake. Similarly, Antonelli et al. (2020) [25] identified PC (18:3/18:2), PC (18:2/18:2), PC (34:3), LPG (0:0/16:0), LPG (0:0/16:1), and several SQDGs in inflorescences of *Cannabis sativa* during a comprehensive polar lipidome study using high-resolution mass spectrometry and cheminformatics. Even though Cerrato et al. (2021) [15] outlined the analytical workflow for polar lipid characterization, to the best of our knowledge no studies have been done on polar lipids with phospholipid characterization of the hemp seed hulls, a major by-product of hemp seed obtained during the dehulling process. In this regard, we examined the polar lipid fractions derived from both hemp seed hulls and hemp cake using UHPLC/HRMS and NMR.

The residual biomass of hemp seed by-products recovered after the removal of oil (hexane extract) was further extracted with EtOH at 65 °C and partitioned into water- and chloroform-soluble fractions by liquid/liquid extraction. Due to their lipophilicities, polar lipids were concentrated in the chloroform fraction, and then separated into neutral, glycolipid, and phospholipid subfractions by silica gel-based SPE, as described by Ryckebosch et al. (2012) [26]. The ^1^H NMR signals belonging to the glycerol backbone (5.26, 4.29 and 4.13 ppm) and fatty acid acyl chain (5.23, 2.78, 2.30, 2.04, 1.60, 1.28, 0.97 and 0.88 ppm) of neutral lipid fractions eluted with chloroform derived from both hemp by-products strongly suggested the presence of predominantly TAGs [17,27]. We already reported the characterization of TAGs in hemp oil derived from both hemp seed and hemp by-products, and thus no further action was taken on the chloroform-eluted SPE neutral lipid fractions [17].

The acetone and methanol fractions were further studied for their lipid components by ^1^H NMR and UHPLC/HRMS analysis using electrospray ionization (ESI) in both positive and negative mode. The glycolipid fraction eluted with acetone from both hemp seed hulls and hemp cake showed identical ^1^H NMR signals belonging to fatty acid moieties and conjugated alkanes (6.20–7.50 ppm) and multiple oxygenated methylene and methane protons (3.00–4.50 ppm) (Figure 1 and Appendix A). The TICs of the acetone fractions of both hemp by-products were almost identical, except that a peak eluted at 0.5–1.0 min in positive mode was absent in hemp seed hulls (Figure 2 and Appendix A). Predominant peaks eluted between 6.50 and 7.50 min belonged to phytyl compounds with molecular mass at *m*/*z* 871, 885, 887, 901, 903 Da etc. were identical with previously reported phytyl derivatives [28]. The presence of significant amounts of molecules other than lipids in the acetone fraction was also supported by the fact that the acetone fraction of both hemp seed hulls and hemp cake showed significantly lower levels of total fatty acids i.e., <330 mg/g in hemp seed hulls. The acetone fraction of hemp cake had only 148 mg/g total fatty acids whereas chloroform and methanol fractions had >642 mg/g total fatty acids (Table 4). DAGs were eluted between 6.31 and 7.00 min and nine DAGs were identified based on the UHPLC/HRMS. ESI is a soft ionization technique widely used for lipid analysis, generating intact molecular ions typically observed as ammonium, sodium, or proton adducts in positive mode [29,30,31].

In the current study, we observed ammonium adduct ions for all DAGs that were chosen from a LipidMAPS database search [32]. The neutral loss of fatty acid (FA) acyl chains in positive mode allowed the identification of FA acyl chains attached to the glycerol backbone of individual DAGs. A representative MS spectrum and fragmentation of DAG with FA acyl side chain 18:3/18:2 is shown in Figure 5. The fragment ions at *m*/*z* 337.27381 and 335.25821 were generated by the neutral loss of FA 18:3 and 18:2, respectively from the parent ion at *m*/*z* 615.49697 [M + H]^+^. Accordingly, the structure of the DAG was identified as DAG 18:3/18:2. Nine DAGs were identified, and the heat-map analysis based on the individual peak areas clearly demonstrated that DAG with acyl side chain 18:2/18:2 was the dominant one, followed by DAG with acyl side chain 18:3/18:2 (Table 2). The position of individual fatty acyl chains within the glycerol backbone was not determined. No SQDGs or other GL were detected in hemp seed by-products in the glycolipid fraction eluted with acetone, even though SQDGs were reported in the inflorescences of *C. sativa* [25].

The ^1^H NMR signals of the phospholipid fraction eluted with methanol showed signals belonging to a fatty acid moiety (5.36, 2.80, 2.35, 2.09, 1.63, 1.35, 1.00 and 0.92 ppm) and glyceride (5.26, 4.46 and 4.19 ppm), and additional methylene signals (4.30 and 4.03 ppm) strongly suggested the presence of phospholipid [12], which was further confirmed by ^31^P NMR measurement. The ^31^P NMR spectra of the methanol fraction of both hemp cake and hemp seed hulls are shown in Figure 3, and suggested the presence of phosphate groups belonging to PC, 1-LPC, PE and LPE [25]. In addition, ^31^P NMR was used to determine the relative proportion of those phospholipids in the MeOH fraction using the areas of resonance in the ^31^P spectrum (Table 5). Among them, PC was the major phospholipid found in the MeOH fraction of both hemp cake and hemp seed hulls, covering 49.3% and 37.1% of total phospholipid, respectively, and their concentrations were similar to a previous report [23]. UHPLC/HRMS analysis led to the identification of six LPCs, five LPEs, six Pes, and thirteen PCs. The TICs of the methanol fraction of both hemp seed hulls and hemp cake were almost identical, with major compound eluted between 6 to 8 min (Figure 2 and Appendix A). For all phospholipids, i.e., LPCs, LPEs, Pes, or PCs, protonated adducts ions [M + H]^+^ were observed and those ions were selected for lipid search using the LipidMAPS database [32]. The identification of PC and LPC was made based on the strong diagnostic fragment at *m*/*z* 184.0733 in positive mode, which belongs to the protonated phosphocholine head group [30,31]. On the other hand, PE and LPE were identified based on the major fragment [M + H-141.0186 Da]^+^ observed resulting from the loss of the phosphoethanolamine head group in positive mode and the deprotonated fragment at *m*/*z* 140.0188, which belongs to phosphoethanolamine in negative mode [30,31].

The mass spectra of PCs with molecular ions at *m*/*z* 782.56914 and 758.56939, as well as PE with molecular ion at *m*/*z* 716.52186, in positive mode eluted around 6.60 min in the hemp seed hulls MeOH fraction are shown in Figure 6a. The LipidMAPS database search indicated that those three molecular ion peaks belong to PC 36:4, PC 34:2 and PE 34:2, respectively, with accurate masses of less than 3 ppm (Table 3). The diagnostic fragment at *m*/*z* 184 Da observed in positive mode resulting from the MSMS of *m*/*z* 782.56914 and 758.56939 further suggested that they belong to PCs (Figure 6b,c). The mass spectra and fragmentation ions of those three molecules in negative mode are shown in Figure 7. In negative mode [M + COO]^−^ ions at *m*/*z* 826.56132 and *m*/*z* 802.56108 were observed for those PCs. The fragment ion at *m*/*z* 279.23306 of [FA 18:2-H]^−^ peak observed for PC 36:4 confirmed that the two FA acyl chains were 18:2. Similarly, the fragment ions at *m*/*z* 279.23325 for [FA 18:2-H]^−^ and *m*/*z* 255.23300 for [FA 16:0-H]^−^ observed for PC 34:2 confirmed that the FA acid acyl chains were 18:2 and 16:0. The higher abundance of the ion at *m*/*z* 279.23325 versus *m*/*z* 255.23300 in the product ion spectrum suggested that the position of 18:2 should be *sn*-2 [33,34,35]. Moreover, observation of [M–H–FA18:2–CH_3_]^−^ or, [M–H–FA18:2–CH_3_–H_2_O]^−^ or both in PCs suggested the position of FA18:2 is in the *sn*-2 position of the glycerol backbone because the loss of fatty acid at *sn*-2 was reported to be more abundant than the ion arising from the loss of the fatty acid substituted at *sn*-1 [33]. Moreover, in PC, the intensity of the *sn*-2 carboxylate anion i.e., R_2_COO^−^ peak was reported to be higher than the intensity of the *sn*-1 carboxylate peak (R_2_COO^−^) [24]. Similarly, the position of the FA acyl chain in the glycerol backbone of all PCs was assigned (Table 3). The PE 34:2 in positive mode, on the other hand, gave a diagnostic fragment peak at *m*/*z* 575.50402 [M + H-141]^+^, representing the neutral loss of phosphoethanolamine (141 Da) in positive mode. The deprotonated fragment at *m*/*z* 140.01078 belonging to phosphoethanolamine in negative mode confirmed its identification. The fragmentation ions at *m*/*z* 279.23303 for [FA 18:2-H]^−^ and *m*/*z* 255.23310 for [FA 16:0-H]^−^observed for PE 34:2 suggested the FA acid acyl chains to be 18:2 and 16:0. Their peak intensity ratios further suggested that the 18:2 should be in the *sn*-2 position in the glycerol backbone. It has been reported that in PE, the ion intensity ratios of fatty acids [FA-H]^−^*sn*-2/*sn*-1 acyl substituents were greater than 1 in negative mode [35].

LPCs were also detected in the MeOH fraction eluted between 2.42 and 5.00 min (Figure 2). All LPCs showed a diagnostic fragment at *m*/*z* 184 Da in positive mode. A representative mass spectrum and fragmentation of a LPC with molecular adduct ion at *m*/*z* 520.34003 with retention time of 4.00 min is shown in Figure 8. A LipidMAPS database search matched LPC 18:2 with a mass accuracy of less than 3 ppm, and the diagnostic fragment at *m*/*z* 184.07323 confirmed that the molecular ion *m*/*z* 520.34003 belonged to an LPC [24]. The significantly higher intensity peak of ion at *m*/*z* 104.10729 compared to peak ion at *m*/*z* 124.9994 indicated that the FA acyl chain was in the *sn*-1 position. This is because it has been reported that a difference of over 30-fold in the peak intensity ratio of product ions at *m*/*z* 104 and 147 corresponds to a sodiated *sn*-1-acyllysophatidycholine in comparison to *sn*-2- acyllysophatidycholine [36]. In the present study, we observed a protonated cyclic fragment at *m*/*z* 124 Da, instead of sodium adducts ion at *m*/*z* 147 Da reported in the literature.

Five LPEs were also identified in the SPE phospholipid fraction and were eluted between 3.53 and 4.82 min (Figure 2). Mass spectra and fragment ions of LPE 18:2 observed in both positive and negative mode are shown in Figure 9. The diagnostic fragment peak resulting from the neutral loss of 141 Da at *m*/*z* 337.27400 i.e., [M + H-141]^+^ in positive mode suggested it belongs to LPE. The loss of fatty acid 18:2 resulting in an ion at *m*/*z* 279.23306 in negative mode confirmed the acyl side chain to be linoleic acid. Accordingly, the structure was determined as LPE 18:2. All LPEs detected either in hemp cake or in hemp seed hulls had a strong diagnostic product ion peak of PE class i.e., [M + H-141]^+^, suggesting *sn*-1 position of the FA in the glycerol backbone. Lee et al. (2011) [37] well documented that 1-LPE (FA acyl group at *sn*-1 position) produces [M + H-141]^+^ ion while the 2-LPE (FA acyl group at *sn*-2 position) produces an intense product ion due to water loss. Moreover, the intensity of the ion at *m*/*z* 196 Da was significantly higher than the ion at *m*/*z* 214 Da (Figure 9d) in all LPEs in negative mode, which also supported the conclusion that the FA acyl chain was in the *sn*-1 position because the relative intensity of these product ions, i.e., the ratio of 196/214 is greater than 1 for the LPC *sn*-1 regioisomer and less than 1 for *sn*-2 regioisomer [38]. Besides LPCs, LPEs, Pes, and PSs, a few unknown phospholipid signals were also detected in the MeOH fraction and were eluted between 5.68 and 5.89 min (Figure 2). They had molecular ions at *m*/*z* 812.54358, 790.55963 and 814.55988. Those molecules had diagnostic fragment *m*/*z* 184 Da in positive mode, but they could not be matched to any PC in the LipidMAPS database. Representative MS spectra of 814.55988 and fragmentation in both positive and negative mode are shown in Appendix A. Buré et al. (2016) [24] also reported lysophosphatidylinositol (LPI), phosphatidic acid (PA), phosphatidylglycerol (PG), phosphatidylserine (PS) and cardiolipin (CL) in hemp cake besides LPE, PE, LPC and PC. Even though we characterized the majority of the polar lipids, including DAG, PL, and phytyls, present in the EtOH extract of residual biomass of hemp cake and hemp seed hulls, no reasonable amount of LPI, PA, PG, PS or CL was detected in polar lipid fractions derived from EtOH extract, suggesting either the EtOH was not a good solvent to extract all PL or the concentration of LPI, PA, PG, PS and CL were lower due to difference in hemp variety. The ^31^P NMR spectrum of hemp by-products also confirmed the presence of LPEs, PEs, LPCs and PCs as major the group of PL in hemp seed hulls and hemp cake except two unknown groups of phospholipids present in hemp seed hulls (Figure 3).

The phospholipid fraction eluted with MeOH derived from hemp cake was further subjected to HPLC purification and the two major compounds were isolated. The HPLC chromatogram shown in Appendix A clearly demonstrates the presence of multiple compounds in the MeOH fraction. A small amount of two major phospholipids eluted at 15.4 and 17.10 min were purified for the first time from hemp seed by-products by HPLC and were identified as 1,2-dilinoleoylphosphatidylcholine (PC 18:2/18:2) and 1-palmitoyl-2-linoleoylphosphatidylcholine (PC 16:0/18:2) based on in-depth spectral analyses including 1D- and 2D-NMR and HRMS spectra.

## 4. Conclusions

We extracted and analyzed polar lipids from hemp seed by-products. To the best of our knowledge, this is the first report of the identification of nine DAGs, six LPCs, five LPEs, eight Pes, and thirteen PCs from hemp seed hulls, even though their presence in hemp cake was reported previously. There has been no previous report on the characterization of DAGs in hemp cake and hemp seed hulls. ESI source was used for ionization of polar lipid analysis and ^31^P NMR study confirmed the presence of phospholipids in the targeted SPE fraction. The LPCs, LPEs, PCs and PEs were identified by HRMS analysis based on the diagnostic fragment obtained in both positive and negative modes. The fragmentation pattern and intensities of diagnostic fragments allowed us to identify the position of the fatty acyl chain in the glycerol backbone for LPC, LPE, PC and PE. DAG 18:2/18:2, LPC 18:2, LPE 18:2, PE 18:2/18:2 and PC 18:2/18:2 were the predominant molecules within their classes determined based on the heat-map analysis. Fatty acid analysis of SPE fractions indicated that linoleic acid was the major fatty acid present within polar lipid fractions ranging from 41.1% to 57.2% of the total fatty acids. HPLC purification of MeOH fractions led to the isolation of two major PCs, and their structure were determined to be 1,2-dilinoleoylphosphatidylcholine and 1-palmitoyl-2-dilinoleoylphosphatidylcholine based on spectral analyses, including NMR and HRMS.

## 5. Materials and Methods

### 5.1. General

The ^1^H NMR spectra were measured on a Bruker 700 MHz spectrometer using a 5 mm cryogenically cooled probe. The ^31^P spectra were acquired on a Bruker 500 MHz spectrometer using a 5 mm BBFO probe tuned to 202.46 MHz. The polar lipid extracts were prepared in a blend of CDCl_3_/MeOH/CsEDTA(aq) according to Monakhova et. al. (2018) [39]. Semi-preparative HPLC was carried out on an Agilent 1200 Series HPLC (Santa Clara, CA, USA) equipped with a diode array detector. High resolution mass spectra were acquired using a Thermo Fisher Scientific (Waltham, MA, USA) Q Exactive^TM^ Hybrid Quadrupole-Orbitrap^TM^ Mass Spectrometer. GC-FID analysis was carried out on an Agilent Technologies 7890A GC spectrometer (Santa Clara, CA, USA).

### 5.2. Research Material

Research material hemp hulls and hemp cake were provided by Hemp Oil Canada (Ste. Agathe, MB, Canada). Hemp seed by-products and residual biomass after oil extraction were stored at room temperature. The EtOH extract and the fractions thereafter were stored in the freezer (−20 °C) before UHPLC/HRMS (Waltham, MA, USA) analysis and HPLC purification.

### 5.3. Polar Lipid Extraction, Fractionation, and Solid-Phase Extraction (SPE)

The hemp seed hulls (4.0 kg) were extracted twice with hexane percolating at room temperature overnight (12 L and 8 L) and the remaining residual biomass was further extracted with EtOH (8 L) at 60 °C for 2 hours under water bath. The EtOH extract was then concentrated using a rotary evaporator till dry. The concentrated EtOH extract suspended in Milli-Q water (500 mL) and was extracted with chloroform (500 mL × 2). The chloroform-soluble part was dried under vacuum, yielding 64.1 g. Under identical extraction conditions, 58.4 g of chloroform soluble fraction was obtained from hemp cake.

The chloroform fractions of both hemp seed hulls and hemp cake were subjected to solid-phase extraction to separate three major lipid fractions as described previously by Ryckebosch et al. 2011 [26]. Briefly, an SPE column (Discovery DSC-Si Tube 3 mL 0.5–10 g, Sigma-Aldrich, USA or Supelclean^TM^ ENVI^TM^-Carb SPE Tube 6 mL, Supleco, USA) was conditioned with 10 mL chloroform. The chloroform-soluble part of the EtOH extract (200–400 mg) in chloroform (1.0–2.0 mL) was applied to the column. The column was then eluted successfully with chloroform (10 mL), acetone (10 mL) and methanol (10 mL), yielding chloroform, acetone and MeOH fractions. The fractions were evaporated under nitrogen and dried overnight under vacuum, and the percentage of each fraction based on the applied sample is described in Table 1. The ^1^H NMR spectra of SPE fractions were recorded in deuterated solvents.

### 5.4. UHPLC/HRMS Analysis

UHPLC/HRMS data was acquired on an UltiMate 3000 UHPLC system coupled to an Q-Exactive^TM^ Hybrid Quadrupole Orbitrap^TM^ Mass Spectrometer (Thermo Fisher Scientific, Waltham, MA, USA) equipped with a HESI-II probe for electrospray ionization (ESI). Separation was achieved on a Thermo Hypersil Gold C8 column (100 × 2.1 mm, 1.9 μm) at 40 °C. Through a flow-splitter, approximately 1/15 of LC eluent was sent to the mass spectrometer. A makeup solution consisting of 5 mM ammonium formate in IPA/de-ionized/methanol 1/2/7 (*v*/*v*) was delivered constantly at 100 µL/min to MS. The solvent system comprised (A) 10 mM ammonium acetate pH 5 and (B) methanol. The initial gradient was 70% B for the first 0.25 min, which increased linearly to 100% B from 0.25 to 5 min, held at 100% B for 2.5 min at a flow-rate of 500 µL/min.

MS data were acquired in both positive and negative polarities. In each polarity, the acquisition alternates between full MS and data dependent MSMS scans, where the three most abundant precursor ions were subjected to MSMS using 30 eV collision energy. The source parameters were set as follows: sheath gas 15, auxiliary gas flow 4, sweep gas 0, spray voltage 2.1 kV, capillary temperature 300 °C, heater temperature 300 °C. Other MS parameters included instrument resolution of 70,000 for full MS and 35,000 for MSMS, with a mass range from *m*/*z* 190 to 2000.

### 5.5. Fatty Acid Analysis

Fatty acid analysis was done according to the AOAC official method 991.39 (AOAC, 2000) with slight modification in triplicate [40]. Briefly, ~10 mg of SPE fraction (chloroform, acetone and MeOH) was placed in a dry 5 mL screw-capped reaction vial and MeOH (1.0 mL) containing 0.1 mg methyl tricosanoate as an internal standard (IS). The mixture was sonicated and 1.5 N NaOH solution in MeOH (0.5 mL) added, blanketed with nitrogen, heated for 5 min at 100 °C, and cooled for 5 min. BF_3_ 14% solution in MeOH (1.0 mL, Sigma-Aldrich, USA) was added, mixed, blanketed with nitrogen, and heated at 100 °C for 30 min. After cooling, the reaction was quenched by the addition of water (0.5 mL) and the FAME extracted with hexane (2.0 mL). Part of the hexane layer (300–600 μL) was transferred to a GC vial for analysis by GC-FID. GC-FID was carried out on an Agilent Technologies 7890A GC spectrometer using an Omegawax 250 fused silica capillary column (30 m × 0.25 mm × 0.25 μm film thicknesses) for fatty acid analysis. Supelco^®^ 37 component FAME mix and PUFA-3 (Supelco, Bellefonte, PA, USA) were used as fatty acid methyl ester standards. The fatty acid content in hemp oil samples was calculated by the following equation and expressed as mg/g sample.
Fatty acid (mg/g) = (A_X_ × W_IS_ × CF_x_/A_IS_ × W_S_ × 1.04) × 1000(1)
where A_X_ = area counts of fatty acid methyl ester; A_IS_ = area counts of internal standard (methyl tricosanoate); CF_X_ = theoretical detector correlation factor is 1; W_IS_ = weight of IS added to sample in mg; W_S_ = sample mass in mg; and 1.04 is factor necessary to express result as mg fatty acid/g sample.

### 5.6. HPLC Purification of Major PCs

The MeOH fraction of SPE containing primarily phospholipid was dissolved in MeOH (~5 mg/mL) and subjected to HPLC separation after filtering through a 0.20 μm 13 mm nylon membrane syringe filter (VWR, Radnor, PA, USA). The semi-preparative HPLC was performed on an Agilent 1200 series HPLC using Zorbax SB-C18 column (5 μm, 9.4 × 50 mm, Agilent Technologies, USA) under gradient condition with mobile phase 0.1% TFA in H_2_O/CH_3_CN (85:15–5:95 in 0–29 min) with UV detection at 205 nm, and 1,2-dilinoleoylphosphatidylcholine (1.3 mg) and 1-palmitoyl-2-linoleoylphosphatidylcholine (1.5 mg) were purified by multiple injections (100 μL) eluting at 15.48 min and 17.10 min, respectively (Appendix A).

## Figures and Tables

**Figure 1 molecules-27-05856-f001:**
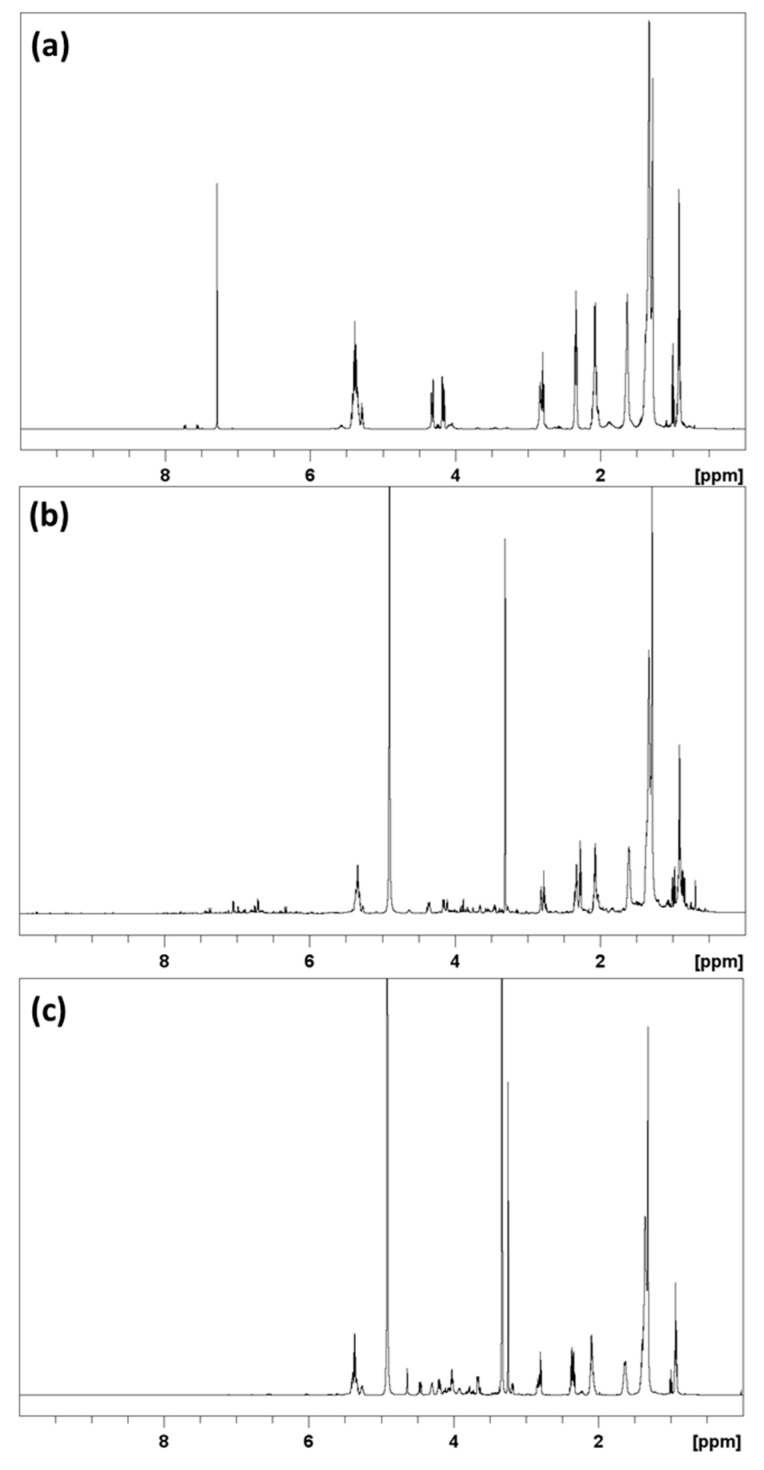
^1^H NMR spectra of hemp seed hull SPE fractions chloroform (**a**) acetone (**b**) and MeOH (**c**). The NMR spectra were measured in CD_3_OD, except for the chloroform fraction, which was measured in CDCl_3_.

**Figure 2 molecules-27-05856-f002:**
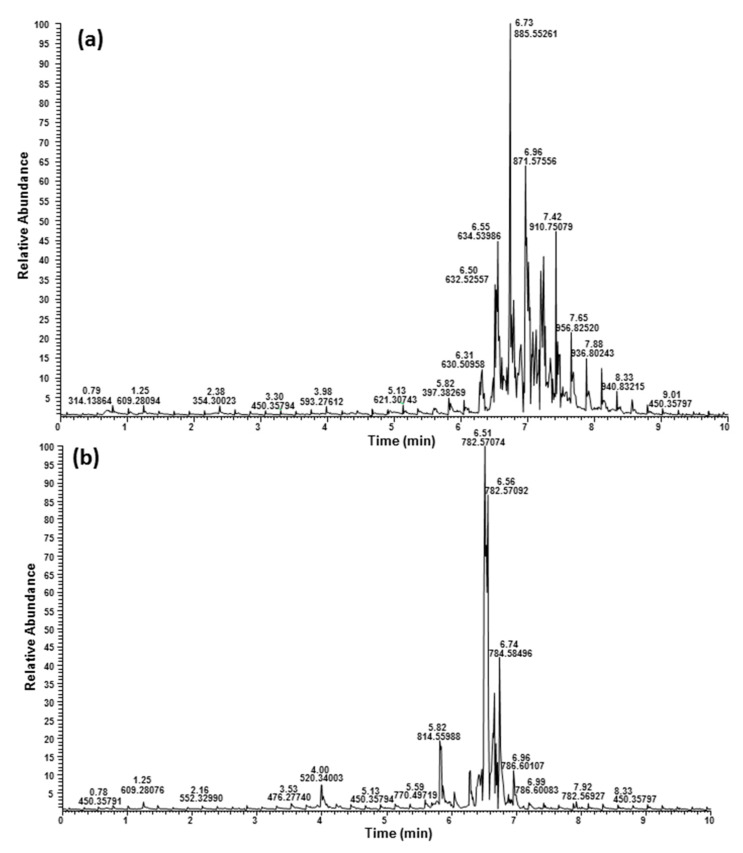
TICs of hemp seed hull SPE fractions acetone (**a**) and MeOH (**b**) in positive mode.

**Figure 3 molecules-27-05856-f003:**
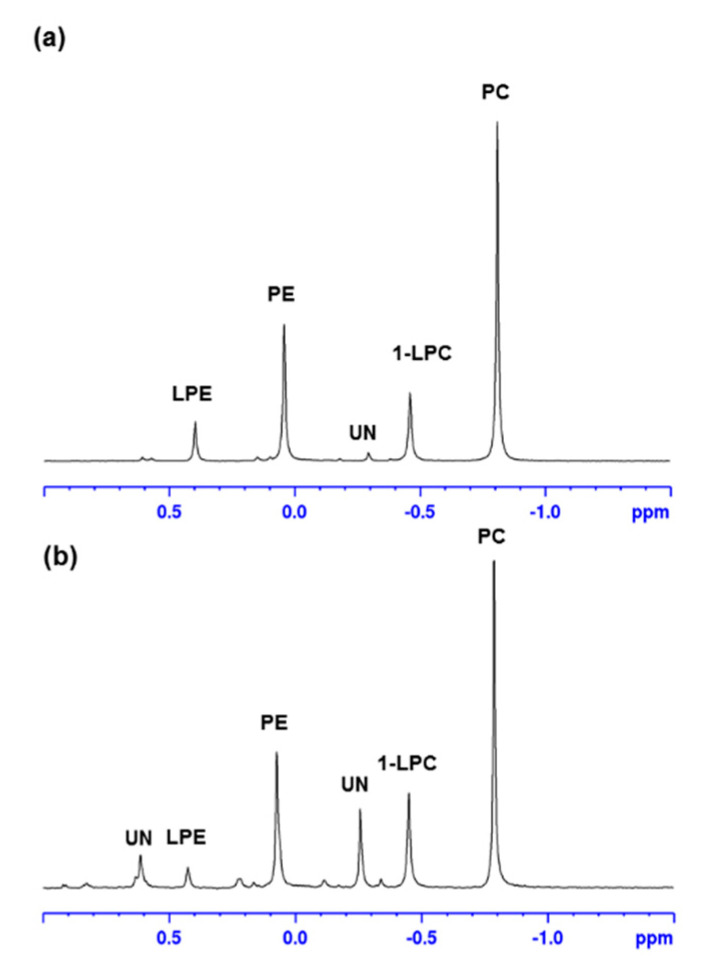
^31^P spectra of MeOH fraction of hemp cake (**a**) and hemp seed hulls (**b**) collected from SPE. LPC—lysophosphatidylcholine, PC—phosphatidylcholine, LPE—lysophosphatidylethanolamine, PE—lysophosphatidylethanolamines, UN—unknown.

**Figure 4 molecules-27-05856-f004:**
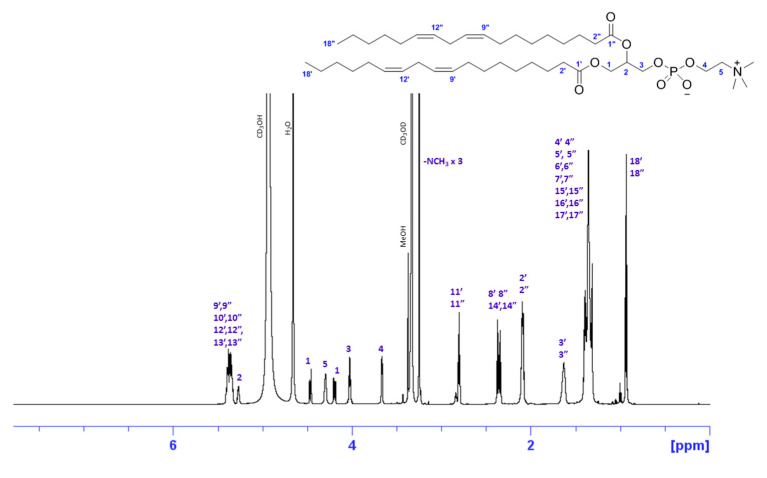
^1^H NMR spectrum of major PC 1,2-dilinoleoylphosphatidylcholine measured in CD_3_OD and assignments of the proton spectrum were made based on 2D-NMR analysis. HRMS *m*/*z* 782.56801 [calculated for C_44_H_81_NO_8_P (M + H)^+^ 782.56943].

**Figure 5 molecules-27-05856-f005:**
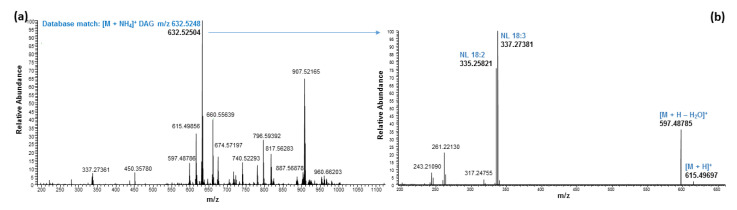
Mass spectra of DAG with molecular ion at *m*/*z* 632.52504 in positive mode eluted around 6.47 min (**a**) and fragmentation ions corresponding to a neutral loss- NL (**b**).

**Figure 6 molecules-27-05856-f006:**
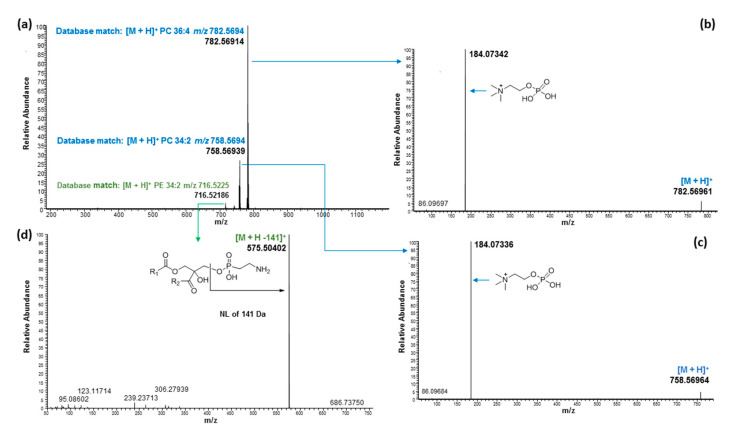
Mass spectra of PCs and PE eluted around 6.60 min with molecular ion at *m*/*z* 782.56914, 758.56939 and 716.52186 in positive mode (**a**), fragmentation of PC with molecular ion *m*/*z* 782.76914 (**b**), fragmentation of PC with molecular ion *m*/*z* 758.56939 (**c**), and fragmentation of PE with molecular ion *m*/*z* 716.52186 (**d**).

**Figure 7 molecules-27-05856-f007:**
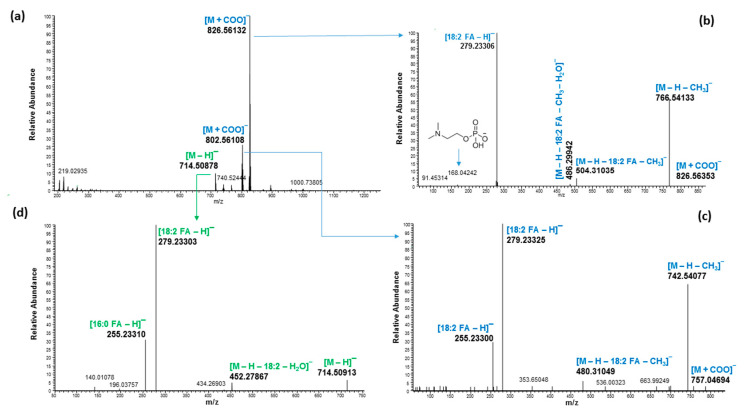
Mass spectra of PCs and PE eluted around 6.60 min with molecular ion at *m*/*z* 826.56132, 802.56108 and 714.50878 in negative mode (**a**) fragmentation of PC with molecular ion *m*/*z* 826.56132 (**b**), fragmentation of PC with molecular ion *m*/*z* 802.56108 (**c**), and fragmentation of PE with molecular ion *m*/*z* 714.50878 (**d**).

**Figure 8 molecules-27-05856-f008:**
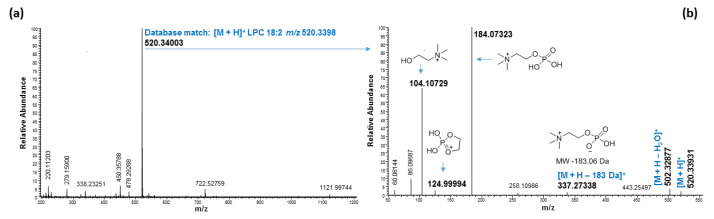
Mass spectrum of LPC eluted at 4.00 min with molecular ion at *m*/*z* 520.34003 in positive mode (**a**) fragmentation of LPC with molecular ion at *m*/*z* 520.34003 (**b**).

**Figure 9 molecules-27-05856-f009:**
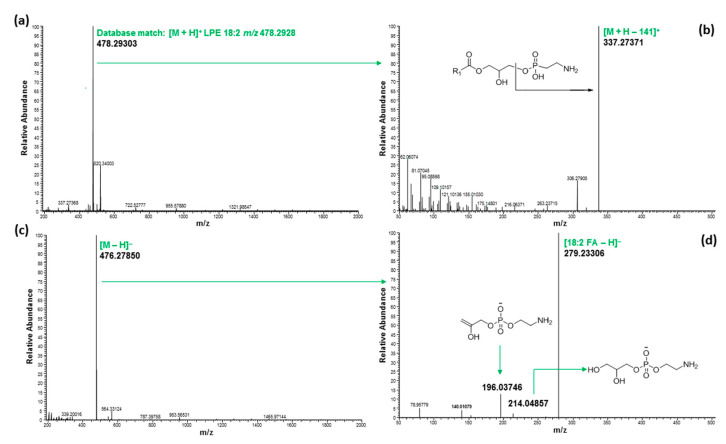
Mass spectrum of LPE eluted at 3.95 min with molecular ion at *m*/*z* 478.29303 in positive mode (**a**), fragmentation of LPC with molecular ion at *m*/*z* 478.29303 (**b**), mass spectrum of LPE eluted at 3.95 min with molecular ion at *m*/*z* 476.27850 in negative mode (**c**), fragmentation of LPC with molecular ion at *m*/*z* 476.27850 (**d**).

**Table 1 molecules-27-05856-t001:** Polar lipid fraction yield and solid phase extraction (SPE) of EtOH extract chloroform soluble fraction of hemp cake and hemp seed hulls.

Sample/Fraction	Hemp Cake—Yield (%)	Hemp Seed Hulls—Yield (%)
EtOH extract-CHCl_3_ soluble part ^1^	1.5	1.6
SPE-Chloroform Fraction ^2^	51.3	63.4
SPE-Acetone Fraction ^2^	22.1	24.7
SPE-Methanol Fraction ^2^	26.4	11.3

^1^ The percentage yield was calculated based on the original hemp by-product biomass. ^2^ The percentage was within EtOH extract-chloroform-soluble fraction.

**Table 2 molecules-27-05856-t002:** Heat map of diacylglycerols (DAGs) identified in the polar lipid fraction (SPE-acetone fraction) of hemp cake (HSCA) and hemp seed hulls (HSHU).

HSCA	HSHU	RT (min)	Measured(*m*/*z*)	Calculated(*m*/*z*)	Error(ppm)	Name:DB ^1^	Formula	Ion	Fatty Acid Identity
		6.31	630.50958	630.50920	0.60	DAG 36:6	C_39_H_64_O_5_	[M + NH_4_]^+^	18:3/18:3
		6.47	632.52521	632.52485	0.57	DAG 36:5	C_39_H_66_O_5_	[M + NH_4_]^+^	18:3/18:2
		6.58	660.55634	660.55615	0.29	DAG 38:5	C_41_H_70_O_5_	[M + NH_4_]^+^	18:3/20:2
		6.68	634.54065	634.54050	0.24	DAG 36:4	C_39_H_68_O_5_	[M + NH_4_]^+^	182:18:2, 18:3/18:1 ^2^
		6.81	636.55658	636.55615	0.67	DAG 36:3	C_39_H_70_O_5_	[M + NH_4_]^+^	18:2/18:1, 18:3/18:0 ^2^
		7.00	638.57248	638.57180	1.06	DAG 36:2	C_39_H_72_O_5_	[M + NH_4_]^+^	18:2/18:0; 18:1/18:1

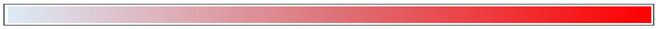
 Low abundance→High abundance ^1^ Search results from LipidMAPS database (https://www.lipidmaps.org (accessed on 7 September 2022)). ^2^ Minor component based on the intensity of fragment ion observed for fatty acyl group.

**Table 3 molecules-27-05856-t003:** Heat map of LPCs, LPEs, Pes, and PCs identified in polar lipid fraction (SPE-MeOH fraction) of hemp cake (HSCA) and hemp seed hulls (HSHU).

HSCA	HSHU	RT (min)	Measured (*m*/*z*)	Calculated (*m*/*z*)	Error (ppm)	Name: DB ^1^	Formula	Ion	FA Identity (R1/R2)
Lysophosphatidylcholines (LPCs)
		2.42	552.32977	552.32960	0.32	LPS 20:1	C_26_H_50_NO_9_P	[M + H]^+^	20:1/0:0
		3.35	518.32256	518.32412	−3.00	LPC 18:3	C_26_H_48_NO_7_P	[M + H]^+^	18:3/0:0
		4.00	520.34003	520.33977	0.51	LPC 18:2	C_26_H_50_NO_7_P	[M + H]^+^	18:2/0:0
		4.20	496.33982	496.33977	0.11	LPC 16:0	C_24_H_50_NO_7_P	[M + H]^+^	16:0/0:0
		4.50	522.35555	522.35542	0.26	LPC 18:1	C_26_H_52_NO_7_P	[M + H]^+^	18:1/0:0
		5.00	524.37082	524.37107	−0.47	LPC 18:0	C_26_H_54_NO_7_P	[M + H]^+^	18:0/0:0
**Lysophosphatidylethanolamines (LPEs)**
		3.53	476.27746	476.27717	0.61	LPE 18:3	C_23_H_42_NO_7_P	[M + H]^+^	18:3/0:0
		3.95	478.29303	478.29281	0.46	LPE 18:2	C_23_H_44_NO_7_P	[M + H]^+^	18:2/0:0
		4.20	454.29294	454.29281	0.29	LPE 16:0	C_21_H_44_NO_7_P	[M + H]^+^	16:0/0:0
		4.40	480.3087	480.30847	0.48	LPE 18:1	C_23_H_46_NO_7_P	[M + H]^+^	18:1/0:0
		4.82	482.3243	482.32411	0.39	LPE 18:0	C_23_H_48_NO_7_P	[M + H]^+^	18:0/0:0
**Phosphatidylethanolamines (PEs)**
		6.14	736.49145	736.49118	0.37	PE 36:6	C_41_H_70_NO_8_P	[M + H]^+^	18:3/18:3
		6.28	738.50757	738.50683	1.00	PE 36:5	C_41_H_72_NO_8_P	[M + H]^+^	18:3/18:2
		6.40	740.52307	740.52248	0.80	PE 36:4	C_41_H_74_NO_8_P	[M + H]^+^	18:2/18:2
		6.49	716.52250	716.52248	0.03	PE 34:2	C_39_H_74_NO_8_P	[M + H]^+^	16:0/18:2
		6.51	742.53979	742.53813	2.23	PE 36:3	C_41_H_76_NO_8_P	[M + H]^+^	18:1/18:2; 18:0/18:3 ^2^
		6.70	744.55438	744.55378	0.80	PE 36:2	C_41_H_78_NO_8_P	[M + H]^+^	18:0/18:2; 18:1/18:1 ^2^
**Phosphatidylcholines (PCs)**
		6.28	778.53900	778.53813	1.12	PC 36:6	C_44_H_76_NO_8_P	[M + H]^+^	18:3/18:3
		6.45	780.55459	780.55378	1.04	PC 36:5	C_44_H_78_NO_8_P	[M + H]^+^	18:3/18:2
		6.60	782.57114	782.56943	2.18	PC 36:4	C_44_H_80_NO_8_P	[M + H]^+^	18:2/18:2; 18:3/18:1 ^2^
		6.60	756.55577	756.55378	2.63	PC 34:3	C_42_H_78_NO_8_P	[M + H]^+^	16:0/18:3
		6.66	758.57012	758.56943	0.91	PC 34:2	C_42_H_80_NO_8_P	[M + H]^+^	16:0/18:2
		6.78	784.58642	784.58508	1.71	PC 36:3	C_44_H_82_NO_8_P	[M + H]^+^	18:0/18:3; 18:1/18:2
		6.95	786.60112	786.60073	0.49	PC 36:2	C_44_H_84_NO_8_P	[M + H]^+^	18:0/18:2; 18:1/18:1
		6.96	760.58553	760.58508	0.59	PC 34:1	C_42_H_82_NO_8_P	[M + H]^+^	16:0/18:1
		7.00	812.61623	812.61638	−0.19	PC 38:3	C_46_H_86_NO_8_P	[M + H]^+^	20:0/18:3
		7.20	814.63265	814.63203	0.76	PC 38:2	C_46_H_88_NO_8_P	[M + H]^+^	20:0/18:2

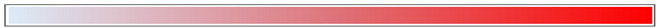
 Low abundance→High abundance, ^1^ Search results from LipidMAPS database (https://www.lipidmaps.org (accessed on 7 September 2022)). ^2^ Minor component based on the intensity of fragment ion observed for fatty acyl group.

**Table 4 molecules-27-05856-t004:** Fatty acid profile of chloroform, acetone and MeOH fractions after SPE of hemp cake (HSCA) and hemp seed hulls (HSHU). Results are expressed in mg/g fraction.

Fatty Acid (FA)	HSCA-CHCl_3_	HSCA-Acetone	HSCA-MeOH	HSHU-CHCl_3_	HSHU-Acetone	HSHU-MeOH
Myristic acid (C14:0)	1.2 ± 0.2 (0.2)	1.7 ± 0.1 (1.6)	1.4 ± 0.7 (0.2)	0.4 ± 0.7 (0.0)	0.6 ± 0.5 (0.2)	1.0 ± 0.2 (0.2)
Myristoleic acid (C14:1)	-	2.5 ± 0.2 (1.7)	-	-	1.1 ± 0.2 (0.5)	
Pentadecanoic acid (C15:0)	-	-	-	-	-	-
*cis*-10-Pentadecenoic acid (C15:1)	-	-	-	-	-	-
Palmitic acid (C16:0)	67.6 ± 1.7 (10.1)	23.1 ± 2.4 (15.6)	99.7 ± 4.3 (15.5)	72.2 ± 10.2 (8.8)	48.8 ± 0.9 (14.7)	111.9 ± 2.1 (18.6)
Palmitoleic acid (C16:1 *n*-7)	1.0 ± 0.1 (0.2)	1.0 ± 0.3 (0.7)	1.2 ± 0.3 (0.2)	2.1 ± 1.5 (0.3)	10.8 ± 0.6 (3.3)	1.1 ± 0.2 (0.2)
C16:2 *n*-4	-	-	-	-	0.9 ± 0.8 (0.3)	-
C17:0 (Heptadecanoic acid)	0.9 ± 0.0 (0.1)	-	1.4 ± 0.1 (0.2)	-	-	1.4 ± 0.0 (0.2)
C16:3 *n*-4	-	-	-	-	-	-
*cis*-10-Heptadecenoic acid (C17:1)	-	-	-	-	-	-
C16:4 *n*-1	-	-	-	-	-	-
Stearic acid (C18:0)	19.5 ± 0.5 (2.9)	4.9 ± 0.7 (3.3)	27.7 ± 0.7 (4.0)	21.2 ± 2.1 (2.6)	11.5 ± 0.2 (3.5)	22.4 ± 0.4 (3.7)
Oleic acid (C18:1 *n*-9)	47.8 ± 0.6 (7.1)	8.3 ± 2.0 (5.6)	20.9 ± 0.5 (3.3)	81.5 ± 13.9 (9.9)	34.3 ± 0.7 (10.4)	26.2 ± 1.2 (4.4)
*cis*-Vaccenic acid (C18:1 *n*-7)	7.1 ± 0.1 (1.1)	2.3 ± 0.4 (1.6)	11.6 ± 0.3 (1.8)	-	-	12.5 ± 0.6 (2.1)
Linoleic acid (C18:2 *n*-6)	355.1 ± 4.8 (53.0)	60.9 ± 10.3 (41.1)	367.1 ± 8.7 (57.2)	432.8 ± 99.1 (52.2)	136.5 ± 3.0 (41.3)	325.0 ± 10.5 (54.1)
*γ*-linolenic acid (C18:3 *n*-6)	21.7 ± 0.2 (3.2)	3.7 ± 0.7 (2.5)	12.6 ± 0.3 (2.0)	0.5 ± 0.9 (0.1)	49.0 ± 1.0 (4.8)	11.2 ± 0.4 (1.9)
*α*-linolenic acid (C18:3 *n*-3)	126.6 ± 1.7 (18.9)	31.1 ± 4.0 (21.0)	87.2 ± 3.4 (13.6)	134.7 ± 36.8 (16.3)	4.0 ± 0.1 (1.2)	74.6 ± 3.4 (12.4)
Stearidonic acid (C18:4 *n*-3)	7.8 ± 0.1 (1.2)	1.5 ± 0.3 (1.0)	2.7 ± 0.0 (0.4)	10.2 ± 2.4 (1.2)	4.1 ± 0.1 (1.2)	1.8 ± 0.1 (0.3)
Arachidic acid (C20:0)	6.4 ± 0.2 (0.9)	1.7 ± 0.2 (1.2)	3.8 ± 0.2 (0.6)	6.9 ± 2.8 (0.8)	1.4 ± 1.4 (0.4)	4.6 ± 0.2 (0.8)
*cis*-11-Eicosenoic acid (C20:1 *n*-9)	2.9 ± 0.0 (0.4)	-	2.4 ± 0.1 (0.4)	4.4 ± 0.4 (0.5)	-	3.0 ± 0.1 (0.5)
*cis*-11,14-Eicosadienoic acid (C20:2)	1.0 ± 0.0 (0.1)	-	1.7 ± 0.1 (0.3)	-	-	1.8 ± 0.1 (0.3)
*cis*-8,11,14-Eicosatrienoic acid (C20:3 *n*-6)	-	-	-	-	-	-
Henicosanoic acid (C21:0)	-	-	-	0.9 ± 1.6 (0.1)	-	-
*cis*-8,11,14-Eicosatrienoic acid (C20:3 *n*-3)	-	-	-	-	-	-
Arachidonic acid (C20:4 *n*-6)	-	-	-	-	-	-
Eicosapentaenoic acid (C20:5 *n*-3)	-	-	-	-	-	-
Behenic acid (C22:0)	3.4 ± 0.1 (0.5)	1.5 ± 0.2 (1.0)	1.5 ± 0.1 (0.2)	4.3 ± 0.0 (0.1)	2.4 ± 1.6 (0.6)	2.2 ± 0.2 (0.4)
Erucic acid (C22:1 *n*-9)	-	1.5 ± 0.1 (1.0)	1.3 ± 0.1 (0.2)	-	-	-
Docosadienoic acid (C22:2 *n*-3)	-	-	-	-	-	-
Lignoceric acid (C24:0)	-	-	-	2.1 ± 0.0 (0.2)	1.9 ± 1.6 (0.6)	-
Docosahexaenoic acid (C22:6 *n*-3)	-	-	-	-	-	-
Others	-	-	-	15.5 ± 10.7 (1.9)	10.5 ± 3.3 (3.2)	-
Total	670.1 (100)	148.1 (100)	642.2 (100)	824.7 (100)	330.2 (100)	600.8 (100)
∑ SFA	99.1 (14.8)	36.8 (24.4)	134.7 (21.0)	107.7 (13.1)	70.5 (21.3)	143.5 (23.9)
∑ MUFA	58.8 (8.8)	14.2 (9.6)	36.2 (5.6)	85.9 (10.4)	47.7 (14.4)	42.9 (7.1)
∑ PUFA	512.2 (76.4)	97.1 (65.5)	471.4 (73.4)	615.6 (74.4)	201.6 (61.0)	414.4 (69.0)

(-) Not detected.

**Table 5 molecules-27-05856-t005:** Percentage proportion of phospholipids in MeOH fraction collected from SPE as determined by ^31^P NMR.

PL-Class	% PL Fraction—HSCA	% PL Fraction—HSHU
Lysophosphatidylcholines (LPC)	14.0	14.9
Phosphatidylcholines (PC)	49.3	37.1
Lysophosphatidylethanolamines (LPE)	6.8	3.3
Phosphatidylethanolamines (PE)	24.2	20.6
Unknown	<1.0	10.1

HSCA—hemp cake, HSHU—hemp seed hulls.

## Data Availability

The data presented in this study are available on request from the corresponding author.

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
