# Peer review of "Analysis of Polar Lipids in Hemp (*Cannabis sativa* L.) By-Products by Ultra-High Performance Liquid Chromatography and High-Resolution Mass Spectrometry"

_molecules, 2022, doi:10.3390/molecules27185856_

Round 1

Reviewer 1 Report

The polar lipids in hemp by-products were analyzed by ultra-high performance liquid chromatography and high-resolution mass spectrometry. UPLC/HRMS analyses of remaining polar lipids lead to the identification of nine diacylglycerols, six lyso-phosphatidylcholines, five lyso-phosphatidylethanolamines, eight phosphatidylethanolamines and thirteen phosphatidylcholines for the first time from hemp seed hulls. It is innovative to a certain extent. However, in order to improve the quality of the manuscript, a major revision of the current manuscript is recommended.

1. In 4.3, the author didn’t mention the amount of hexane used; In the extraction process, the author only used ethanol (1/2 w/v) to extract once, whether the required substances can be completely extracted; In addition, the properties of the two samples are different (hemp seed hulls and hemp cake), and whether the content of chloroform extract extracted by the same method is credible.

2. In 4.5, GC-FID was used for fatty acid analysis, but the author did not mention it in the abstract and 4.1.

3. In materials and methods, the 1H NMR analysis method of SPE components is not introduced.

4. In 2.3, the title of Figure 2 is inconsistent with the content description. Moreover, Figures S3 and S4 are not sorted according to the contents mentioned in the text.

5. The title of Figure 3 (a, b) didn’t indicate the sample name.

6. The format of references was not uniform, such as whether the journal name is italicized.

7. The authors should provide as clear figures as possible, including supplementary materials.

Author Response

Thank you for your positive comments. The following changes were made to the revised manuscript and we are hopeful that will answer all the questions and suggestions raised.

  1. The volume of hexane used for extraction of oil from both hemp seed by-products is provided in the experimental section 4.3. For simplicity, we also provided the volume of EtOH used for the extraction of polar lipids. EtOH extraction was done only one time in order to reduce the volume of EtOH which has a higher boiling point, and needs more energy and time to evaporate as compared to hexane. To enhance extraction efficiency of EtOH extraction was done at a higher temperature (60⁰C). The hemp seed hulls and hemp cake have distinctive properties specifically in carbohydrate and protein content, their lipid content is in a similar range, which was reported in our earlier research paper (Banskota et al., Molecules, 2022, 27, 4794) and reference is added in the revised text. Chloroform was used for the fractionation of EtOH extract to concentrate the lipid fraction and worked well. Regarding EtOH as a choice of extracting solvent, we have already discussed its relevance (lines 473-479).
  2. Information on GC-FID was added both in the abstract and experimental section 4.1.
  3. A new statement was added in the material and methods section describing 1H NMR measurement of SPE fractions.
  4. The legend of Figures 2, S3 and S4 were revised according to the content described in the text. We appreciate your suggestion.
  5. In Figure 3 samples (a, b) were identified in the legend.
  6. The references were reformatted according to the journal’s guidelines i.e., all journals’ names are italicized.
  7. Figures both in the main text and in the supplementary document were revised by changing the figure size and format JPG file to a TIFF file.

Reviewer 2 Report

The article is very interesting and well organized. It is extension of previous work for the authors; in-depth identification of polar lipids in hemps. However, many revisions should be performed before final acceptance

1-     Introduction is very concise and informative. However, many related research papers should be cited  e.g.

https://www.mdpi.com/1420-3049/27/10/3358

https://www.sciencedirect.com/science/article/pii/S0268005X22002752

Untargeted characterization of extracts from Cannabis sativa L. cultivars by gas and liquid chromatography coupled to mass spectrometry in high resolution mode - ScienceDirect

https://www.tandfonline.com/doi/abs/10.1080/87559129.2019.1600539?journalCode=lfri20

https://link.springer.com/protocol/10.1007/978-1-0716-1410-5_17

https://link.springer.com/article/10.1007/s00216-019-02247-6

2-     furthermore, the new findings and novelty over the last aforementioned article should be highlighted in the discussion

3-     Resolution of fig 1  and fig 5 should be improved

4-     Lines 370-386 should be under separate title “ conclusion “ . it is very informative and precise thanks

Best of luck

Author Response

We appreciate your comments and positive feedback. The following changes were made to the revised manuscript.

  1. Additional statements were added in the introduction including relevant references describing the biological properties and the latest report on the lipid analysis of hemp.
  2. Discussion section was also revised accordingly adding a new statement describing the latest research finding on lipid analysis of hemp seed oils.
  3. Resolution of all figures were changed to improve the quality of the figures including Figure 1 and 5.
  4. Separate conclusion section was created with the statement previously described in lines 370-386.

Round 2

Reviewer 2 Report

the authors did all the recommendations , thanks.

the paper could be published in the current form